# Transforming psychology education to include global mental health

Brigitte Khoury[1]  and Viviane De Castro Pecanha[2]

[1]Department of Psychiatry, American University of Beirut, Beirut, Lebanon and [2]Department of International Psychology, The Chicago School of Professional Psychology, International Psychology, Online Campus, Chicago, IL, USA

global mental health; psychology; education; social justice; human rights

**Author for correspondence:**
Brigitte Khoury,
Email: bk03@aub.edu.lb

## Abstract

In recent years, the reality of global migration has brought the lack of understanding of mental health needs across different cultures into sharp focus. Psychology programs are not up to date on global issues and are often experienced as inadequate in preparing graduates to meet the challenges of society today. The field of education and training in psychology has hardly evolved since the last two decades. On the other hand, the mental health needs arising locally and globally require a knowledge base and a set of skills future psychologists need to have in order to be able to work and grow professionally. In addition, most psychologists in the western world are bound, at some point in their career, to be in contact with immigrants or refugees to offer them services and be a source of support for such a vulnerable population. Also, the field of psychology is witnessing more movement among psychologists than ever before, whereby many professionals move to another country, to work, volunteer, gain or provide training, consult and much more. This requires a certain level of preparation, which psychologists need to be aware of and ready to engage in before and after they move. This article highlights different psychology programs around the world that include global mental health in their programs. It discusses essential aspects and skills that psychologists need to learn to be prepared to work globally with various populations and to expand their skills beyond service providing to more management and policy work. Topics such as human rights and social justice, advocacy, health management and policymaking are addressed as important competencies to be gained during the education and training of future psychologists.

## Impact statement

With the globalization of the mental health field, psychology programs are lagging behind addressing in their curricula global events impacting the mental health of people and societies worldwide. This results in inadequacy in preparing professionals to meet the diverse challenges of society today. This article highlights the inclusion of global mental health in the training and education of psychology. Future psychologists should be trained on subjects including human rights and social justice, advocacy, health management, policymaking and leadership. These topics are treated as important knowledge and competencies to acquire for the 21st century psychologist. By developing these skills, future psychologists may apply their expertise to advance the global mental health field improving services (access and equity) to benefit communities located in various parts of the globe.

## Introduction

The historical events occurring in the 21st century such as natural disasters (earthquakes, hurricanes and tsunamis), man-made disasters (wars, conflicts and terrorist attacks) and epidemic and pandemic outbreaks (COVID-19, Ebola, H1N1) confirm the need to equip psychologists and general healthcare workers to provide evidence-based, culturally relevant and contextually appropriate mental health interventions focused on human rights and social justice frameworks across the world (WHO, 2019; 2022; Bahar et al., 2021).

Integrating global mental healthcare into undergraduate and graduate academic programs is crucial. Specifically in areas that require mental health professionals to work with underrepresented minorities and at risk groups from various backgrounds – migrants, neglected populations, individuals enduring extreme trauma, survivors of man-made, natural disasters, and human rights violations as well as living under the poverty line and low, middle-resource communities (Murray et al., 2014; WHO, 2019; 2022; Meffert, 2021). The inclusion of global mental health (GMH) disciplines and/or programs into the academic curricula would also aid in developing psychologists and healthcare professionals' skills. Among these skills, advocate for social inclusion, propose and implement policies working in collaboration with governments and

organizations, coordinate capacity building initiatives, adhere to international ethical principles in the field of psychology and facilitate the development of sustainable holistic treatments taking into account important contextual determinants (e.g., historical events, economics, politics, environment, globalization and technology) to promote health and mental health (IUPsyS, 2008; Patel and Prince, 2010; de Castro and McGrath, 2015; Kuruvilla et al., 2018; Weir, 2020).

## Global mental health

In addition to the historical events mentioned above, the global threats to mental health indicated in the World Health Organization's (WHO, 2022) report are, "economic downturns and social polarization; public health emergencies; widespread humanitarian emergencies and forced displacement; and the growing climate crisis" (p. 26). There are three bases of evidence supporting the inclusion of GMH in the curricula of healthcare-related disciplines (Rajabzadeh et al., 2021) and particularly in psychology (de Castro and McGrath, 2015). First, the significant number of cross-cultural and transnational research discussing inequities in the provision of mental health care and the barriers to access mental health services in high-, middle- and low-income countries (Patel and Prince, 2010; Patel et al., 2014). Second, the exponential increase in epidemiological research confirming the burden of mental disorders in all world regions prior and after the COVID-19 pandemic (Rehm and Shield, 2019; Nochaiwong et al., 2021). In fact, according to the WHO (2019), "people with mental disorders experience disproportionately higher rates of disability and mortality" (p. 2). Third, the evidence that there are effective pharmacological and psychological interventions available for the treatment of mental health disorders, and that general healthcare workers can prescribe the medications and deliver the services (Patel and Thornicroft, 2009).

The advancements of the GMH field can be traced back to 2007 when the Lancet Medical Journal published a series of articles focusing on the topic (Patel and Prince, 2010). In the mentioned series, an action plan was drafted with the goal of scaling up services based on human rights approaches and empirical studies to treat individuals diagnosed with mental health disorders. In the decades that followed Lancet's call for action, the field of GMH gained prominent visibility prompting the interest of psychologists and healthcare professionals in diverse parts of the world. GMH became an area of study and research supporting practices concerned with inclusion, equity, human rights, and social justice and responsibility (Koplan et al., 2009; Civitelli et al., 2020; Meffert, 2021).

## GMH education

The WHO along with a commission of organizations and individuals devoted to enhancing the mental health care across nations launched the Mental Health Global Action Program (mhGAP) and the Movement for Global Mental Health. The WHO has acknowledged the mhGAP to be its "flagship" (p. 2) program in mental health. The program provides evidence-based guidelines for the general healthcare professionals to treat specific mental and neurological disorders in multiple settings (Patel and Prince, 2010). The conditions prioritized in the mhGAP program and assessed through the International Statistical Classification of Diseases and Related Health Problems, Eleventh revision (ICD-11; WHO, 2019) are depression, psychoses, bipolar affective disorder, schizophrenia, alcohol and substance disorders, suicidal ideation, anxiety disorders, epilepsy, dementia,

intellectual disabilities, as well as mental and behavioral disorders observed during childhood and adolescence, and due to the exposure to acute stress. While the WHO was proposing the mhGAP program, a coalition of institutions and individuals was established to develop The Movement for Global Mental Health program. The Movement for Global Mental Health program was inspired by the success of the HIV Treatment Action Campaign – their mission was to expand the access to psychopharmacological and mental health treatment in various parts of the world.

Heeding the worldwide call to move toward a more inclusive and effective service model guiding GMH practices, and using the examples set by Doctors Without Borders (*Médecins Sans Frontières)* and Partners in Health (Hixon et al., 2013), various schools in the healthcare field (e.g., medicine, psychology, social work, nursing and public health) decided to develop disciplines and graduate programs in the area. The academic interest in GMH has been on the rise especially in the USA and other developed nations. The Centre for Global Mental Health in London focusing on capacity building initiatives in GMH, the Master's in International Mental Health Policy, Services and Research offered by the University of Lisbon, the Grand Challenge in Global Mental Health led by the National Institute for Mental Health and the Global Alliance for Chronic Diseases (Patel and Prince, 2010), the Global Mental Health Programs at McGill University, the Columbia University Global Mental Health programs, Johns Hopkins Global Mental Health program housed within the School of Public Health, School of Medicine Yale Global Mental Health program and the Master's in Arts and the Doctorate programs in International Psychology offered by The Chicago School of Professional Psychology, training psychologists and other healthcare professionals to design and evaluate interventions in the GMH field based on human rights and social justice frameworks are illustrated (de Castro and McGrath, 2015).

The Caracas Declaration signed in 1990 was one of the most important hallmarks to advance mental health services in Latin America and the Caribbean (LAC). With the support of the Pan American Health Organization (PAHO), LAC was tasked to build a collaborative and interprofessional model of assistance integrating mental health into the community and the primary healthcare system (Mascayano et al., 2021). Based on a research coordinated by PAHO (2017), the prevalence of mental, neurological and substance use disorders in the Americas ranged from 19% to 24%, and 4% in Latin America. However, it is important to note that the report focused on 12-month mental health illness prevalence studies from "the most representative community-based survey of a country" (Kohn et al., 2018, p. 2). In addition, studies reporting the prevalence of mental health disorders in Latin American countries are scarce and they vary considerably depending on the design and methodology used. Therefore, the data presented in the PAHO (2017) report may not reflect the reality of mental health disorders in Latin America. The numbers may be even higher. Despite these obstacles, the mental health services provided in LAC significantly improved in the past 20 years incorporating mental health in primary care and enhancing the access of mental health care in the community (Caldas de Almeida, 2013; Sapag et al., 2021). In Brazil, the Federal University of São Paulo (Department of Psychiatry) offers opportunities, at the master and doctorate levels, to conduct research studies in the field of GMH (Jacob et al., 2007; Sharan et al., 2009; Thornicroft et al., 2011; Oquendo et al., 2018; Marques et al., 2022). Nonetheless, formal training and graduate programs in GMH, encouraging culturally and contextually oriented knowledge, competencies, skills and

practices, are yet to be developed in South America (Rich et al., 2018; Rich et al., 2020).

## Competencies in the GMH field

Around the world, the discipline of psychology is discussed, studied, adapted and practiced within a multitude of contexts based on diverse social, education, political, economic and legal norms. The changes that are being made in the profession itself to address the never-ending challenges of global societies (mental health illness) reflect the development of academic programs, disciplines and training to equip the professionals with international core and specific competencies to conduct ethical work that is inclusive and based on cultural/contextual practices (de Castro and McGrath, 2015; Morgan-Consoli et al., 2018; Stevens et al., 2018).

Countries differ vastly on the models of education and training of psychologists (i.e., number of years, theoretical frameworks, foundational literature, concepts, pedagogy, format of delivery, competencies, learning outcomes and internship requirement). Considering the broad scope, in 2013 a task force committee was assembled to identify universal benchmark competencies in the field of professional psychology during the 5th International Congress on Licensure, Certification and Credentialing. The conference held in Sweden, Stockholm was by invitation-only and included 75 participants from 18 countries (e.g., Norway, United Kingdom, Romania, South Africa, China, New Zealand, Colombia, Canada and USA to name a few) representing five continents. The event counted with the presence of important and prestigious global associations in the field of psychology – the International Association of Applied Psychology, the International Test Commission and the International Union of Psychological Science (IUPsyS) to support the development of a collaborative project with the goal to produce a document proposing an international competence model. Participants had access to relevant sources and material presenting the current available models on competence for psychologists prior to the first meeting and ran as an extended sequence of workshops consisting of lengthy small group discussions.

Planners and participants also agreed to broaden discussions beyond local, regional and national differences to generate a mutual understanding of psychological competencies. At the end of the event, the task force committee proposed an agreement to establish a multi-stakeholder international project – The International Project on Competence in Psychology (International Association of Applied Psychology and International Union of Psychological Science, 2016).

To serve as the basis for a global professional identity in the field of professional psychology, and potentially be recognized as a universal system to assist with program accreditation, professional credentialing and regulation of professional competence and conduct, the International Declaration of Core Competencies in Professional Psychology (IDCCPP) is currently known as a foundational document guiding the work of psychologists worldwide. IDCCPP includes a preamble with working definitions of commonly used terms to have a comprehensive list of the language used within the declaration. It then lists the competencies that may be both explicitly applied worldwide (ethical practices) and less explicit but equally important (cultural humility). Following the definitions, the document presents the clusters of competencies based on psychological knowledge and skills underpinning the core competencies as well as competencies relating to professional behaviors and activities (Von Treuer and Reynolds, 2017). The competencies proposed are not specific but rather general, encouraging organizations, communities and regions to accommodate their local contexts as needed. Further, IDCCPP acknowledges that adaptations and translations may lead to variations in educational and training requirements across cultures and believe this will capture the variety of richness and expression in the international professional psychology field.

Recognizing the importance of training professionals working in the healthcare field, Buzza et al. (2018) conducted a research to identify relevant GMH domains and competencies required to provide effective services in diverse parts of the world. Interestingly, several domains introduced in the study are closely aligned with the IDCCPP clusters and competencies (Table 1). Culturally informed and contextually driven psychologists, and healthcare workers in

**Table 1.** Examples excerpt from global mental health (GMH) competencies for fellowship training and International Declaration of Core Competencies in Professional Psychology (IDCCP)

| GMH domains | IDCCPP clusters/competencies |
|---|---|
| Structural determinants of mental health | Works with knowledge and understanding of the historical, political, social and cultural context of clients, colleagues and relevant others |
| Cultural aspects of mental health | Works with diversity and demonstrates cultural competence – e.g., cultural humility |
| Understanding the health systems and resources in mental health | Possesses specialized psychological knowledge and skills – e.g., psychological concepts, constructs, theories, methods, practice and research methodology |
| Psychotherapy | Conducts psychological interventions with individuals, groups, communities, organizations, systems or society |
| Training and education | Possesses psychological knowledge and skills Operates as an evidence-based practitioner and acts professionally – e.g., maintain competence as a psychologist |
| Clinical supervision | Reflects on own work – e.g., validates reflections with peers or supervisors |
| Interprofessionalism and leadership | Relates appropriately to clients and others – e.g., establishes, maintains and develops appropriate working relationships with colleagues in psychology and other professions |
| Health equity and ethics | Practices ethically – e.g., applies relevant ethics code, adhere to laws and rules of practice, resolve ethical dilemmas |
| Quality improvement | Reflects on own work – e.g., implements areas for improvement in one's practice |
| Self-reflection and care | Reflects on own work – e.g., the impact of one's own values and beliefs have on one's practice |

GMH, play significant roles contributing to enhance social inclusiveness, advocate for practices focused on social justice and advance policies protecting human rights. Yet, balancing competing needs (e.g., competence vs. cultural and social practices) remains a challenge to expand the training in GMH (Khoury et al., 2022). As the mental health crisis continues worldwide, the field of GMH will need to increase the number of competent professionals meeting the requirements to work in the mental health area and ready to deliver mental health services in international contexts (Weir, 2020; Sapag et al., 2021; WHO, 2022). Psychologists will need to reinvent themselves stepping outside of the comfort zone of their own cultural frameworks and practices to ultimately develop a flexible, humble and adaptative global identity in order to provide effective services.

## Humans rights, social justice and advocacy

Psychologists and general healthcare workers in the GMH field may encounter situations involving unethical, unfair, bias and discriminatory practices (Asanbe et al., 2018; Patel et al., 2018; Millum et al., 2019). In order to offer transformative mental health services, professionals should engage in critical self-reflection (e.g., one's own impact on others), nurture deep respect for cultures (e.g., cultural humility) and demonstrate a strong ethical commitment, following the ethical principles included in the Universal Declaration of Ethical Principles for Psychologists, implemented by IUPsyS, in 2008, as an ethical framework guiding the work of psychologists in international contexts (Gauthier et al., 2010).

Studies have shown that worldwide, many individuals with mental illness are isolated from society, and they are institutionalized for long periods of time with inhuman living conditions (Thornicroft, 2006; Bonnie and Zelle, 2019; Millum et al., 2019). They are also commonly forced into treatment without their consent and without access to protect their rights. Some are even denied their civil and political rights and are discriminated against when it comes to citizenship, education, employment, transportation, housing and access to health care and the legal system. WHO considers human rights violations of people with disabilities (physical and intellectual) to be an emergency in global health that needs attention. Widespread human rights abuses are still observed in mental health institutes, hospitals and in the communities even after the 1960's deinstitutionalization movement (Bonnie and Zelle, 2019; WHO, 2019; 2022).

In response to the abuses and violations, some countries have started to integrate health, mental health and human rights into medical education. To illustrate, The Health Professions Council of South Africa has placed a mandatory human rights course in higher education institutions (London et al., 2007) after the Truth and Reconciliation Commission had identified that several healthcare professionals participated in cruel and unethical practices with their patients, despite their commitment (Hippocratic Oath) to safeguard their lives and to condemn methods violating human rights. Creating health equity and adhering to ethical principles are competencies expected and highlighted in the GMH domains (Buzza et al., 2018), in the ICDDP (International Association of Applied Psychology and International Union of Psychological Science, 2016) and the UDEPP (IUPsyS, 2008). In fact, psychologists and healthcare professionals working toward health equity, in its broad sense (including mental health), are advocating for the right for equal access to health and mental health services, on behalf of underrepresented, vulnerable and disadvantaged individuals,

people and communities – e.g., low-income societies, refugees, displaced, asylum seekers, immigrants, homeless, racial, and ethnic minorities, LGBTQ+, individuals living with disabilities, chronic and debilitating (life-threatening) disorders, statelessness persons, among many other groups. As noted in the WHO's (2022) report, deficient infrastructure and poor access to mental health care are significant structural risk factors associated with the development of mental health disorders, whereas social justice, inclusion, integration and human rights-based care are considered protective structural determinants.

Successful transformational change in GMH requires creating innovative curricula in psychology and other related fields based on human rights, social justice and advocacy models. This movement is already observed in the fields of medicine psychology, social work, nursing and public health, as several graduate programs (discussed in the GMH education section) were launched integrating human rights, social justice and advocacy in GMH. Embedding GMH into the academic curricula is a social responsibility (WHO, 2022). Psychologists and professionals in the GMH field can and should propose innovative solutions to address the disparities contributing to the GMH crisis, promote the determinants of health and mental health, protecting the rights of people with mental health disorders, engage in advocacy and social justice initiatives to demystify mental health disorders and ensure that all are treated with respect and dignity, encourage commitment and empowerment and combine efforts participating in interprofessional projects promoting capacity building to combat stigma and discrimination. The need to train new and senior health specialists in the area of GMH throughout their careers has increased and will continue to grow exponentially especially considering the ongoing challenges around the world, and the WHO comprehensive mental health action plan proposed for the next 17 years to invest in transformational GMH (Bakshi et al., 2015; Weir, 2020; Rajabzadeh et al., 2021; WHO, 2022).

## Teaching health policy and health management

In 2001, the WHO released a report on World Health that focuses on mental health. It suggested several solutions to the problems of GMH including establishing national policies and legislation. In fact, the WHO's (2022) report put forward the inseparable links between mental health and public health, specifically human rights and socioeconomic development. This indicates that the transfiguration of policy and practice in mental health has a positive impact on individuals, societies and countries at large. Thus, contributing to mental health can improve the lives of everyone. The literature has shown that the ideal approach to building capacity in global mental healthcare will require partnerships amid professional resources in high-income countries and encouraging health organizations in low- and middle-income countries (LMICs) (Fricchione et al., 2012). The result of these partnerships will be sustainable academic relations that can instruct a new generation of primary care physicians, psychologists and others, ultimately, health specialists. Researchers claim that it is a crucial educational component to teach health policy and practice, in training and to promote better intervention, prevention and strategies. Sola et al. (2019) proved in her article that by looking at health policy through the lens of academia, trainees were able to develop a new perception on how health policy activities can contribute to teaching, publications and leadership opportunities. When health policy was incorporated into the teaching, students were better situated to integrate health

policy skills in their scholastic and professional careers as well (Sola et al., 2019).

In the past 20 years, there has been increasing overlap between the agendas of the government and the department of health, and the aims of health psychologists in many regions in the world. This has often included exploring factors behind behaviors and evaluating interventions established and studied that aim to change health behaviors for individuals, groups and the community (Abraham and Michie, 2005). This most notably may lead to additional funds for healthcare services, as interventions and strategies for disease prevention and healthcare maintenance would develop and become more efficient, reducing sickness costs. At the community level, these strategies would also lead to more public engagement and by extension trust in the healthcare system due to greater and more comprehensive knowledge and awareness of health. Such a process may then lead to an improved level of health in the general community and to project developments aimed at developing higher quality and more preventative services.

The synchronous systems model, a theoretical and clinical approach that defines and includes individual differences in people and settings, could support healthcare services and influence policy. Within this model's scope, cost effectiveness and medical efficacy may improve and become more efficient, suggesting psychologists may play a central role in the triage of medical services and healthcare policies. Gregerson (1995, p. 1) explains: "The Synchronous Systems Model provides theory, supportive data, and clinical assessment devices to strengthen clinical psychology's role in medical settings."

Further models since then have also been developed offering more opportunities for psychologists to develop their roles in treatment. For those models to be applied effectively, psychologists must become more interdisciplinary, familiar with the healthcare culture, expand and adjust their skill sets, maintain collaboration with other health discipline as well as have a larger perspective for advocacy (Johnson and Marrero, 2016).

Psychologists have been able to successfully change federal statutes in the USA in the past 20 years. However, some obstacles and limitations have remained clear. In an article published in 1995, DeLeon and colleagues explained that Health Psychology was still very often overlooked at the time and that clinical psychologists' interventions were often prioritized. They continued: "federal health policy decisions, including management of excessive federal health spending will dictate the growth and opportunities for health psychologists" (DeLeon et al., 1995, p. 493). A talk presented at a conference on The Socially Responsible Psychologist – A Symposium in Honor of M. Brewster Smith, held at the University of California, Santa Cruz, on April 16, 1988, was published in a paper in 2021 (Kelman, 2021) again in support of the need for psychologists in policy. It discussed the socially responsible citizen when it came to policymaking and speaking up and the similar concept of the socially responsible psychologist. In line with that article, a responsible citizen is one that brings independent judgment on policy process and assesses policies with an independent view as per their values, with the aim of citizens having a more active say in government and determining policy. These aims intersect with the role of the socially responsible psychologist, but distinctly psychologists may bring an alternative perspective given their specified knowledge and experience in their field, in the evaluation of policy. Therefore, psychologists must meet their "citizen obligations" with the added value of their knowledge to the essential role of changing policies for the needs of the community in order to protect the public (Kelman, 2021).

Finally, Friedman (2006) explains that for efforts to truly be successful, effective and clearly understood, psychologists must have a comprehensive understanding of policy, government and research. For that he suggests more training for graduate students that would include policy-related issues, an increase in interdisciplinary and transdisciplinary training and research and "by integrating the principles and concepts of complex adaptive systems within research and training, and by diversifying research methods to allow careful study of important policy and systems issues" (Friedman, 2006, p. 6).

When looking at the implementation strategy of such programs, it is important to focus on applying hybrid designs for better knowledge regarding the effectiveness of interventions in diverse environments. Such designs and interventions target adaptability in different areas, task division, supervision and training, along with fidelity checks and looking into retention rates and follow-up with participants (Meffert et al., 2016).

Recently, the limitations are also being addressed as some universities opened a department for health policy and management with courses dedicated to this field. This department does offer courses that are available for students that are in psychology graduate programs; however, they are electives. For example, in the University of South Hampton, UK an international social policy course exists within the program of Health Psychology. This course exposes students to policymaking and planning, yet not specific to health policy. On the other hand, other universities such as Northwestern University, USA offer a master's degree in health psychology that includes a specific mental health policy elective. This pattern also appears in several other programs in Europe, United Kingdom and the USA.

## Teaching leadership skills

Teaching leadership skills in graduate psychology programs is also necessary since psychologists tend to find themselves serving leadership positions in the community that they are not always prepared for. Based on the APA ethical code of conduct, psychologists are dedicated to integrity, justice, fidelity and responsibility in practice, as well as the use of psychological knowledge for the benefit of individuals and society (American Psychological Association, 2010). Teaching leadership skills during higher education can support the future implementation of psychologists' knowledge in an informative way. Psychologists in training can be influenced by learning the following fundamental leadership qualities: self-awareness, congruence, commitment, cooperation, shared purpose and handling conflict with civility (Kois et al., 2016). Leadership activities enhance student training by developing professional identities, and organizational and communication skills that are vital for service provision and advancement, as well as professionally with colleagues. Various psychology doctoral programs, internships and postdoctoral residency guidelines provide students with leadership roles; however, there are still no explicit requirements for acquiring such experience. Clinical psychology training programs and mentorship beyond sole academic guidance are central to psychology students' journey to leadership development. This also allows for opportunities for student involvement in departmental policies as well as learning modeled leadership behaviors from faculty mentors (Kois et al., 2016). Nonetheless, it is important to note that access to education remains a barrier in the 21st century. In a study conducted by the United Nations Educational, Scientific and Cultural Organization (UNESCO), high tuition rate was identified as one of

the most important obstacles to achieve universal access to higher education. Provision of funds (private and governmental) is crucial to address this problem supporting the training of competent professionals especially in LMICs (UNESCO, 2020).

## Conclusion

It is necessary to acknowledge that the internationalization of such a curriculum faces barriers that slow the process down, particularly regarding the educators' time limitations, enthusiasm and other commitments that might get in the way of developing an international psychology course (Stevens and McGrath, 2017). Educators are encouraged to advocate for the internalization of the psychology curriculum, as well as communicate its importance to students, faculty and affiliated institutions. The goal of international psychology courses is to familiarize students with non-Western research and phenomena while pushing them to critically assess how global events affect psychosocial experiences.

Psychology education in the 21st century must provide psychologists in training the knowledge and skills to fulfill the needs of individuals and communities worldwide. Widening the curricula scope to include social justice, human rights, advocacy and health policy and management will be preparing the new generation of psychologists to be GMH professionals who can offer their services and expertise wherever they are needed, with cultural humility and respect making them role models for other health professions.

**Open peer review.** To view the open peer review materials for this article, please visit http://doi.org/10.1017/gmh.2023.11.

**Author contribution.** B.K. is the main contributor, and V.D.C.P. has contributed significantly to the development of the manuscript.

**Competing interest.** The authors declare no conflicts of interest.

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
