## [Reviewer Report]

August 1st, 2022

Dear Editor of the Cambridge Prism series,

I am happy to submit this manuscript titled: “Transforming Psychology Education to Include Global Mental Health” to the journal of Global Mental Health. 

My co-author and I think that this article will be an important addition to the literature on global psychology and education, which will hopefully highlight the future of psychology worldwide with a focus on the preparation of young psychologists in training to become global psychologists.

This manuscript has not been submitted to any other journal and is our original work.

Thank you for all your support

Best,

Brigitte Khoury, PhD

---

## [Reviewer Report]

*Comments to Author*: Expand the summary to acknowledge the barriers and identify specific steps that can be taken to begin internationalization of curriculum.

(See also Stevens, M. J. & McGrath, B. (2017). A stand-alone course on international psychology. In G. Rich, U. P. Gielen, and H. Takooshian (Eds.). Internationalizing the teaching of psychology. Charlotte, NC: Information Age Publishing.)

---

## [Reviewer Report]

*Comments to Author*: This paper presents a reasonable summary and point worthy of publication. The only adjustment I would make is to shorten several sentences, in particular, the first one which is hard to read due to its length.

Not for a change to your argument, but towards the end of your paper you mention that students should be taught about government and policy etc. I would also teach leadership skills so graduates know how to execute their knowledge in an influential way. (We are trialing the efficacy of teaching leadership skills presently...) Best wishes

---

## [Reviewer Report]

*Comments to Author*: Thank you for this thoughtful manuscript. The reviewers acknowledge the importance of this work and recommend a few minor revisions to strengthen the final version.